# Predictors of Early Response, Flares, and Long-Term Adverse Renal Outcomes in Proliferative Lupus Nephritis: A 100-Month Median Follow-Up of an Inception Cohort

**DOI:** 10.3390/jcm11175017

**Published:** 2022-08-26

**Authors:** Eleni Kapsia, Smaragdi Marinaki, Ioannis Michelakis, George Liapis, Petros P. Sfikakis, John Boletis, Maria G. Tektonidou

**Affiliations:** 1Department of Nephrology and Renal Transplantation, Laiko Hospital, Medical School, National & Kapodistrian University of Athens, 11527 Athens, Greece; 2Department of Hygiene, Epidemiology and Medical Statistics, Medical School, National & Kapodistrian University of Athens, 11527 Athens, Greece; 3Department of Pathology, Medical School, National & Kapodistrian University of Athens, 11527 Athens, Greece; 4Rheumatology Unit, First Department of Propaedeutic Internal Medicine, Joint Academic Rheumatology Program, Laiko Hospital, Medical School, National & Kapodistrian University of Athens, 11527 Athens, Greece

**Keywords:** lupus nephritis, renal response, renal survival, twelve-month proteinuria, interstitial fibrosis/tubular atrophy

## Abstract

**Objective:** To define predictors of response, time to response, flares, and long-term renal outcome in an inception cohort of proliferative lupus nephritis (PLN). **Methods:** We included 100 patients (80% female; mean age 31 ± 13 years) with biopsy-proven PLN (III, IV, III/IV + V). Clinical, laboratory, histological and therapeutical parameters were recorded at baseline, 6, 9, 12, 18, 24, 36, 72 months, time of flare, and last follow-up visit. Logistic and Cox-regression models were applied. **Results:** After induction treatment (69% received cyclophosphamide (CYC) and 27% mycophenolic acid (MPA)), partial (PR) or complete (CR) response was achieved in 59% (26% CR, 33% PR) and 67% (43% CR, 24% PR) of patients at 3 and 6 months, respectively; median time to PR was 3 months (IQR 5) and median time to CR was 6 months (IQR 9). Baseline proteinuria <1.5 g/day correlated with a shorter time to CR (HR 1.77) and with CR at 3, 6, and 9 months (OR 9.4, OR 5.3 and OR 3.7, respectively). During 100-month median follow-up, 33% of patients had ≥1 renal flares (median time: 38 months). Proteinuria >0.8 g/day at 12 months was associated with a higher risk of flares (OR 4.12), while MPA and mixed classes with lower risk (OR 0.14 and OR 0.13, respectively). Baseline proteinuria >2 g/day and 12-month proteinuria >0.8 g/day correlated with a shorter time to flare (HR 2.56 and HR 2.57, respectively). At the end of follow-up, 10% developed stage 3–4 chronic kidney disease (CKD), and 12% end-stage renal disease (ESRD). Twelve-month proteinuria >0.8 g/day (OR 10.8) and interstitial fibrosis/tubular atrophy >25% (OR 7.7) predicted CKD or ESRD at last visit. **Conclusions:** Baseline proteinuria <1.5 g/day predicted time to CR. Twelve-month proteinuria >0.8 g/day correlated with flares (ever) and time to flare and, along with baseline interstitial fibrosis/tubular atrophy >25%, predicted CKD or ESRD at the last visit.

## 1. Introduction

In systemic lupus erythematosus (SLE), renal involvement is the most common severe complication of the disease affecting approximately 20–60% of patients [1] and conferring a high morbidity and mortality risk [2,3,4]. End-stage renal disease (ESRD) still occurs at varying rates among different cohorts of patients with lupus nephritis (LN), despite the revolutionary changes in LN treatment over the past fifty years [5,6]. A wide range of demographic, socioeconomic and disease-related parameters have been associated with LN’s short- and long-term outcomes. Histological class is a major determinant of renal survival in LN, with the proliferative classes having the worst prognosis compared to other classes [5,7].

The main therapeutic goal in patients with LN is the long-term preservation of renal function, preventing and managing comorbidities, and improving disease-related quality of life [8]. These goals can be achieved by inducing and maintaining disease remission and by preventing and timely treating disease flares [9,10]. In this context, recognizing early predictors of renal response, flare and long-term renal survival is paramount in guiding the therapeutic management of LN patients.

The aim of the present study is (a) to examine the response to treatment and short- and long-term renal outcomes in an inception cohort of patients with proliferative LN; (b) to define predictors of response, time to response, flares and long-term adverse renal outcomes.

## 2. Patients and Methods

### 2.1. Study Population

We examined an inception cohort of 100 patients with proliferative LN diagnosed between 1992 and 2021 and followed up at our joint academic center (Nephrology and Rheumatology Units) at Laiko General Hospital of Athens until April 2022. All patients fulfilled the 2019 classification criteria for SLE and had a biopsy-proven diagnosis of proliferative LN (class III, IV, III/IV + V), classified according to the International Society of Nephrology/ Renal Pathology Society (ISN/RPS) 2003 lupus nephritis classification system. Among 100 patients identified with PLN, 28% (28/100) had class III, 47% (47/100) class IV, 9% (9/100) mixed class III + V and 16% (16/100) mixed class IV + V.

### 2.2. Data Collection

We retrospectively reviewed medical charts of patients and recorded clinical, laboratory, histological and therapeutical parameters at the following time points: time of histological diagnosis of LN; 3, 6, 9, 12, 18, 24, 36, 72 months after the diagnosis; time of renal flare (with or without a repeat biopsy), and at last follow-up visit. Patients with inadequate data and less than 6 months of follow-up were excluded.

Data collected included demographic parameters, time from SLE diagnosis to LN onset, disease activity (assessed by Systemic Lupus Erythematosus Disease Activity Index 2000, SLEDAI-2K score) [11], anti-ds DNA titers, C3 and C4 levels, serum urea (Ur), creatinine (Cr) and albumin, eGFR (based on the CKD-EPI formula), 24-h proteinuria, urine sediment, renal biopsy histological parameters (LN class, activity index (AI), chronicity index (CI), crescents, interstitial fibrosis/tubular atrophy (IF/TA)), and treatment regimens.

All data (demographic, clinical, laboratory and histological) were extracted from our anonymized cohort dataset. No individualized or identifiable data are presented in this study. The study was approved by the Institutional Review Board (protocol number 1725/14-12-2017) of Laiko General Hospital of Athens.

### 2.3. Patient and Public Involvement

Patients and/or the public were not involved in this research’s design, conduct, reporting or dissemination plans.

### 2.4. Definitions

Response and flares were defined according to the 2012 EULAR/ERA-EDTA [8] and the 2012 KDIGO recommendations [12] as follows: Active urine sediment: the presence of >5 RBCs/hpf or ≥1 red cell casts. Complete response (CR): proteinuria <500 mg/24 h and serum creatinine within 10% of baseline values. Partial response (PR): ≥50% reduction in proteinuria to subnephrotic levels and serum creatinine within 10% of baseline values. Nephritic flare: increase in glomerular haematuria by ≥10 RBCs/hpf with or without a decrease in eGFR by ≥10%, irrespective of changes in proteinuria. Nephrotic flare: reproducible doubling of proteinuria to >1000 mg/24 h if a complete response had been previously achieved or reproducible doubling of proteinuria to ≥2000 mg/24 h if a partial response has been previously achieved. Stage 3–4 chronic kidney disease (CKD): eGFR 15–60 mL/min/1.73 m^2^ for more than 3 months, and end-stage renal disease (ESRD) as eGFR < 15 mL/min/1.73 m^2^ or initiation of renal replacement therapy [13].

### 2.5. Statistical Analysis

Continuous variables were expressed as the mean value and standard deviation or median value and interquartile range (IQR), whereas categorical variables as frequencies and percentages. To investigate the differences in baseline parameters between patients with different therapeutical schemes, the *t*-test and Mann–Whitney *U* test for independent samples for continuous variables and the χ^2^ and Fisher exact test for categorical variables were applied. Univariate logistic regression analyses were performed to estimate the prognostic effect of various variables on the risk of renal flare and adverse renal outcomes. Cox regression analyses investigated the association between patients’ variables and time to response or response at specific time points. Variables found to be significant in the univariate analyses were included in the multivariate models. In the models identifying predictors of flare and adverse renal outcome, due to a small number of events, selected independent variables were included in the multivariate analyses after multicollinearity was examined. Multicollinearity issues were assessed with appropriate tests (χ^2^, Cramer’s V coefficient (V), analysis of variants (ANOVA), Spearman correlation coefficient (r_s_) and Mann–Whitney *U* test) before performing multivariate analyses. Strong correlation among specific variables excluded some parameters from the multivariate models. Significance was set at α = 0.05. The estimated odds ratios (ORs) and hazard ratios (HRs) of both the univariate and multivariate models and the related p-values are presented. Data were analyzed using Stata 13.0 software (Stata Corporation, College Station, TX). All tests proceeded as two-tailed.

## 3. Results

### 3.1. Baseline Characteristics

We studied 100 patients with biopsy-proven proliferative LN. The baseline demographic, clinical, laboratory and histological characteristics and the therapeutic regimens applied are shown in Table 1. All biopsies contained ≥10 glomeruli, and those performed before 2003 were reassessed based on ISN/RPS 2003 classification system. The median follow-up time was 100 months (IQR = 108).

### 3.2. Immunosuppressive Regimens

Induction treatment consisted of cyclophosphamide (CYC) in 69% (69/100) of patients (in combination with rituximab (RTX) in 12/69), mycophenolic acid (MPA) in 27% (27/100, in combination with RTX in 4/27) and RTX alone in 2 patients. All the above patients received corticosteroids. One patient was treated only with corticosteroids due to non-compliance, while one patient did not receive any immunosuppressive treatment due to progression to ESRD soon after LN diagnosis (Table 1). CYC was given in all 69 patients intravenously, following the NIH regimen in 96% and the ELNT regimen in 4% of cases.

Patients treated with CYC differed significantly in several baseline parameters compared to those treated with MPA (Appendix A). Although baseline eGFR did not differ between the two treatment groups, all patients with eGFR <30 mL/min/1.73 m^2^ were treated with CYC. Half of the patients with class III were treated with CYC and half with MPA (13/26 vs. 13/26). CYC was more often preferred over MPA for patients with class IV (87% (40/46) vs. 13% (6/46)) and for patients with class IV + V (88% (14/16) versus 12% (2/16)). Patients with class III + V received more often MPA than CYC (75% (6/8) versus 25% (2/8)) (data not shown).

Maintenance treatment consisted of MPA in 77% (77/100) of patients, azathioprine (AZA) in 8% (8/100) and CYC in 5% (5/100). Ten patients (10/100, 10%) did not receive any immunosuppressive maintenance treatment (Table 1); 5/10 due to early progression to ESRD, 2/10 who received RTX as induction continued with steroids only, 1/10 due to non-compliance, and 2/10 were lost to follow-up after completing the 6-month induction treatment.

The median duration of treatment was 39 months (IQR = 38) and did not differ between the CYC and MPA groups. Additionally, 76% (76/100) of patients received hydroxychloroquine and 65% (65/100) an ACEi or ARB.

### 3.3. Renal Response and Determinants of Response

For up to 72 months, 59% of patients achieved complete (CR) or partial (PR) response at 3 months (26% CR, 33% PR), 67% at 6 months (43% CR, 24% PR), 75% at 9 months (45% CR, 30% PR), 88% at 12 months (69% CR, 19% PR), and 86% at 18 months (71% CR, 15% PR), with similar percentages at all the following time points (Figure 1).

The median time to PR and CR was 3 months (IQR = 5) and 6 months (IQR = 9), respectively. Time to CR was significantly shorter in patients with class III (median/IQR 3/4) compared to patients with class IV (median/IQR 10/13, *p* <0.001) and those with mixed classes (median/IQR 10/11, *p* = 0.006) in the 6, 9, 12 and 18-month time points (*p* for log-rank = 0.49, *p* for Wilcoxon = 0.02) (Appendix A). There was no difference in time to CR between class IV and mixed classes (*p* = 0.94). Time to response between the two treatment groups (CYC versus MPA) did not differ significantly (Appendix A). Median time to CR was 6 months (IQR = 7.5) in the MPA group and 7 months (IQR = 15) in the CYC group (*p* = 0.09).

Moreover, 7% of patients (7/100) of the cohort did not ever respond to treatment, 6 of whom progressed to ESRD, while one patient had a preserved renal function at 7 months but was lost to follow-up afterward. Two nonresponders, who progressed to ESRD after 4 and 5 months, died at 6 and 7 months, respectively, after LN diagnosis.

In Cox regression analysis, proteinuria <1.5 g/day at LN diagnosis was the only parameter that significantly correlated with a shorter time to complete response (HR 1.77, *p* = 0.01) (Table 2).The two treatment groups were compared when they were used alone or in combination with RTX.

We further compared patients who achieved CR at 12-18-24 months versus those with PR or no response at the respective time points. We found that the only significant predictor of CR was baseline proteinuria <1.5 g/day (OR 16.9, *p* = 0.008 for month 12, OR 5.24, *p* = 0.01 for month 18 and OR 4, *p* = 0.03 for month 24) (Appendix A). Although achievement of CR in earlier time points was less frequent, baseline proteinuria <1.5 g/day was also found to be the only significant predictor of CR at 3 months (OR 9.4, *p* <0.001) and 6 months (OR 5.3, *p* = 0.004), and, along with baseline eGFR > 60 mL/min (OR 4.04, *p* = 0.02), predicted CR at 9 months (OR 3.7, *p* = 0.02) (Appendix A).

### 3.4. Renal Flares and Determinants of Flares

Thirty-three percent (33/100) of patients had ≥1 renal flares in a median time of 38 months (IQR = 43); 30% (10/33) of flares were nephritic, and 70% (23/33) nephrotic. Repeat biopsy was performed in 91% (30/33) of cases, and a class switch was observed in 67% (20/30) of them. In all 30 repeat biopsies, active LN lesions were confirmed. In two of them, a pattern of focal segmental glomerulosclerosis was also described, which may have contributed to the nephrotic clinical presentation of these patients.

In univariate logistic regression analysis, longer time to response was identified as a significant risk factor of flare (OR 1.14, *p* = 0.01 for PR, OR 1.05, *p* = 0.01 for CR) (Table 3). Response (CR or PR) at 3 months (OR 1.05, *p* = 0.9), 6 months (OR 1.25, *p* = 0.66) and 9 months (OR 2.2, *p* = 0.19) after treatment did not significantly affect the risk of flare. Statistical significance was found for lack of response (CR or PR) at 12 (OR 3.8, *p* = 0.04), 18 (OR 4.9 *p* = 0.01) and 24 (OR 6.6, *p* = 0.02) months (Appendix A).

Proteinuria >2 g/day at diagnosis (OR 3, *p* = 0.02), and proteinuria >0.8 g/day at 12 months (OR 3.38, *p* = 0.02) were significantly associated with a higher risk of flare. Conversely, parameters associated with a lower risk of flares were older age (OR 0.95, *p* = 0.02) and treatment with MPA (OR 0.25, *p* = 0.02) compared to treatment with CYC (whether these agents were used alone or in combination with RTX). Mixed classes were also found to reduce the risk of flare (OR 0.21, *p* = 0.02), but this association was significant only when mixed classes were compared to class III and not to class IV (Table 3).

Since proteinuria at diagnosis, as a variable, was found to be strongly associated with induction therapy, histological class, and proteinuria at 12 months, it was not included in the multivariate model for the risk of flare. Accordingly, time to remission was strongly correlated with proteinuria at 12 months and LN class and was not included in the multivariate model (Appendix A).

In the multivariate analysis, proteinuria >0.8 g/day at 12 months was a significant predictor of flare (OR 4.12, *p* = 0.02), while treatment with MPA (OR 0.14, *p* = 0.01) and mixed classes were associated with lower flare risk (OR 0.13, *p* = 0.02) (the latter only when compared to class III) (Table 3).

Baseline proteinuria >2 g/day (HR 2.56, *p* = 0.02) and proteinuria >0.8 g/day at 12 months (HR 2.57, *p* = 0.01) were also found to correlate with shorter time to flare (Figure 2).

### 3.5. Patient Survival

At a median follow-up time of 100 months, four deaths were reported. A 59-year-old and heavy smoker patient died due to lung cancer 8 years after LN diagnosis, during which he was on immunosuppressive treatment (he received CYC as induction treatment followed by MPA for 3 years and AZA, due mainly to extrarenal manifestations, for 5 years). One patient, aged 65, died due to bladder cancer 8 years after LN diagnosis. Interestingly, this patient had never received CYC. Two other patients, aged 30 and 38, died from sepsis (7 and 6 months after LN diagnosis, respectively). Both patients had presented with rapid deterioration of renal function and progressed to ESRD very soon after LN diagnosis.

### 3.6. Renal Survival

At the end of the follow-up period, 10% (10/100) of patients developed stage 3–4 CKD, and 12% (12/100) progressed to ESRD.

Six of 12 (50%) patients who reached ESRD were nonresponders, and in 5 of them, ESRD was developed in the first 9 months. Seven (58%) patients progressed to ESRD after 55–183 months from LN diagnosis. eGFR at the time of diagnosis, renal response, and time until ESRD are shown in Appendix A.

We defined the composite endpoint of stage 3–4 CKD and ESRD as adverse long-term renal outcomes. Of note, all patients who had a decrease of eGFR >25% of baseline values at the last follow-up had reached stage 3–4 CKD or ESRD.

In the univariate logistic regression analysis, adverse long-term renal outcome was significantly associated with baseline eGFR < 60 mL/min/1.73 m^2^ (OR 6, *p* = 0.001), interstitial fibrosis/tubular atrophy >25% (OR 5.44, *p* = 0.007), proteinuria >0.8 g/day at 12 months (OR 9.5, *p* = 0.003) and longer time to response (OR 1.1, *p* = 0.01) (Table 4). The risk for stage 3–4 CKD or ESRD was greater for those not achieving any response after 6 months of treatment (OR 6.4, *p* = 0.001), while response at 3 months did not seem to affect long-term renal outcome significantly (OR 2.2, *p* = 0.11) (Appendix A).

eGFR at diagnosis was strongly correlated with baseline proteinuria, proteinuria at 12 months, and interstitial fibrosis/tubular atrophy. Therefore, it was excluded from the multivariate model. Time to remission was strongly correlated with baseline proteinuria, proteinuria at 12 months, and eGFR at diagnosis and was not included in the multivariate model either (Appendix A).

In multivariate analysis, proteinuria >0.8 g/day at 12 months (OR 10.8, *p* = 0.001) and interstitial fibrosis/tubular atrophy >25% at LN diagnosis (OR 7.7, *p* = 0.01) remained significant predictors of adverse renal outcome (Table 4).

## 4. Discussion

A major therapeutic goal in patients with LN is the long-term preservation of renal function by minimizing chronic active or relapsing disease and, at the same time, avoiding excessive drug toxicity. In this context, we need to identify predictors of renal response, relapse, and long-term adverse renal outcomes to optimize our therapeutic strategies. Short-term predictors can also serve as endpoints in large clinical trials investigating new therapeutic agents [14,15,16,17]. In our study, baseline proteinuria <1.5 g/day was the only significant predictor of time to complete response, while proteinuria >0.8 g/day at 12 months and treatment with CYC (compared to MPA) emerged as strong predictors of renal flare. We also showed that adverse long-term renal outcomes (stage 3–4 CKD, ESRD) could only be predicted by interstitial fibrosis/tubular atrophy >25% in baseline biopsy and proteinuria >0.8 g/day at 12 months.

In our study, response to treatment was achieved early after induction treatment; 59% of patients had a renal response at 3 months (26% CR, 33% PR), 67% at 6 months (43% CR, 24% PR), 88% at 12 months (69%CR, 19% PR) and 88% (71% CR, 17% PR) at 2 years. These results are encouraging compared to previous studies reporting 33–50% CR at 6 months, 49–68% CR at 12 months, and 63% CR at 2 years [9,14,18,19,20,21,22,23,24], although our population was predominantly Caucasian. Response to treatment has been associated with better long-term renal outcomes [25,26,27,28], with some studies reporting better renal survival in patients with CR than those with PR only [19,27]. In contrast, others did not find any significant differences [9]. Baseline renal function, proteinuria, hypertension, histological activity and chronicity indexes have been previously suggested as predictors of renal response [9,18,19,24,29,30,31]. In our study, baseline proteinuria >1.5 g/day emerged as the most significant predictor of early (i.e., at 3, 6, 9 months) and late complete response (i.e., at 12, 18 and 24 months). Time to response has not been fully addressed in previous studies, with some reporting a correlation with improved renal survival [28] and others a predictive role of baseline proteinuria [18,32]. Our study predicted time to CR only by baseline proteinuria >1.5 g/day.

Renal flare is a recognized predictor of progressive CKD and morbidity in LN patients [10,33]. Different studies report flare rates of 25–66% [14,34,35,36], largely depending on follow-up time and definitions of flare. In our study, 33% of patients experienced a relapse in a median time of 38 months. Few data are available regarding factors predictive of flare. Attainment of CR, rather than PR, shorter time to CR and maintenance treatment with MPA (compared to AZA) have been the main determinants of flare occurrence across studies [14,35,36,37]. In our study, although a longer time to either CR or PR correlated with a higher risk of flare, this correlation was significant only for responses at 12, 18 and 24 months. Induction treatment with MPA (compared to CYC) and proteinuria <0.8 g/day at 12 months were significantly associated with a lower risk of flare, with the latter finding adding to the value of 12-month proteinuria as a predictor of renal outcome. Proteinuria at the time of LN diagnosis, apart from affecting time to response, was also found to be a risk factor for flare. The cut-off level that was strongly associated with a higher flare risk was determined at 2 g/day. Baseline proteinuria >2 g/day and 12-month proteinuria >0.8 g/day were also found to correlate with a shorter time to flare. Interestingly, pure proliferative classes conferred a greater risk of flare when compared to mixed classes, but the difference was significant only between class III and mixed classes. Other baseline histological characteristics did not affect the risk of flare. However, studies have shown that the activity index of repeat, instead of initial, biopsy may predict flare occurrence and the time of flare [38,39].

Recent studies report cumulative renal survival rates in proliferative LN of 91, 81, 75 and 66% at 5, 10, 15 and 20 years, respectively, and 5-, 10- and 15-year cumulative incidence of ESRD of 3–11%, 6–19% and 19–25% respectively [40,41,42,43]. In our study, 22% of patients reached the composite renal outcome (stage 3–4 CKD or ESRD) in a median follow-up time of 100 months. Several studies have identified baseline parameters as predictors of long-term renal prognosis, such as creatinine, proteinuria, hypertension, activity and chronicity index [43,44], while others [15,16,18,28,40,44,45] highlight the importance of renal parameters at 12 months as single variables (i.e., proteinuria <0.7 g/day or <0.8 g/day) or as composite endpoint (i.e., renal response). Renal response definition, however, is not uniform across studies. The current study demonstrated that the only variables strongly associated with long-term adverse renal outcomes were proteinuria >0.8 g/day at 12 months and interstitial fibrosis/tubular atrophy >25% at the initial biopsy. Baseline proteinuria did not seem to affect renal survival significantly. Some studies have reported that activity and chronicity index, as well as interstitial fibrosis/tubular atrophy, correlate with a worse renal outcome when evaluated in repeat and not in initial biopsies [38,46]. In our study, interstitial fibrosis/tubular atrophy emerged as the only histologic parameter of the initial biopsy that could predict a worse renal outcome. Interstitial fibrosis at initial biopsy has already been associated with adverse renal outcomes [45,47].These results imply that evaluating distinct components of the activity and chronicity indexes in initial and repeat biopsies may be of greater value in predicting renal response and long-term outcome than relying solely on activity and chronicity indexes. While some studies report that a very early response (i.e., at 3 months) can predict long-term renal outcomes [48,49], we did not find that such an early response significantly affected renal survival. In our study, a response at 6, 12, 18 and 24 months predicted a better renal outcome.

The use of an inception cohort of biopsy-proven proliferative (only) LN patients, the long follow-up time of 100 months, and the uniform approach in the setting of a specialized academic center constitute the main strengths of our study. The study’s limitations are its retrospective design and the inclusion of almost exclusively Caucasian patients.

## 5. Conclusions

The current study highlights the value of baseline proteinuria <1.5 g/day as the only early predictor of time to complete response in patients with proliferative LN. It also demonstrates that induction treatment with mycophenolic acid significantly lowers the risk of renal flare. At the same time, 12-month proteinuria >0.8 g/day correlates significantly with flare occurrence and, along with interstitial fibrosis/tubular atrophy >25% at the initial biopsy, strongly predicts long-term renal outcome.

## Figures and Tables

**Figure 1 jcm-11-05017-f001:**
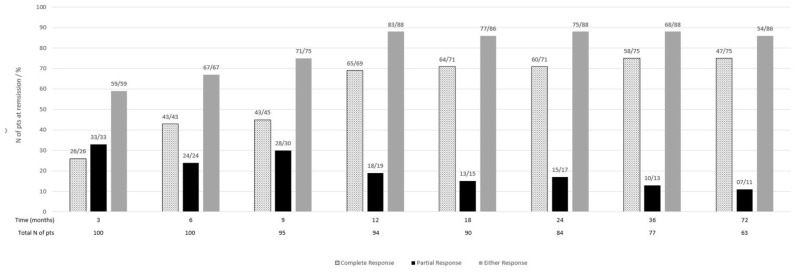
Patients with a renal response at different time points during follow-up.

**Figure 2 jcm-11-05017-f002:**
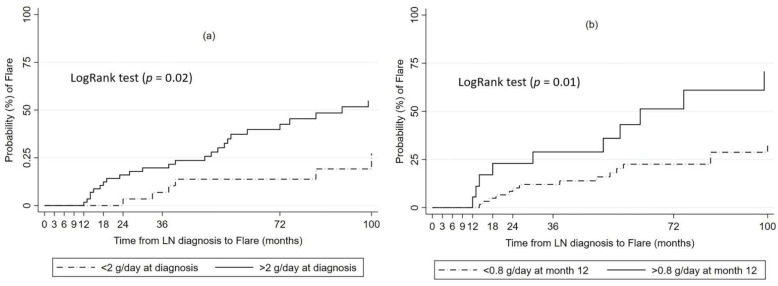
Kaplan–Meier survival estimates of probability for flare according to (**a**) Proteinuria at the time of LN diagnosis, (**b**) Proteinuria at 12 months.

**Table 1 jcm-11-05017-t001:** Baseline demographic, clinical, laboratory and histological characteristics and immunosuppressive treatment regimens.

Baseline Characteristics	Mean ± SD, Median(IQR), N/%
Age (yr) mean ± SD	31 ± 13
Sex (M-F) N/%	20/20–80/80
Race (Caucasian-Other) N/%	96/96–4/4
Time from SLE diagnosis to LN (years)median(IQR)	()
LN as first presentation of SLE N/%	51/51
SLEDAI score median(IQR)	12(4)
Low C3 N/% ^1^	65/77.5
Low C4 N/% ^1^	55/65.5
Positive anti-dsDNA antibodies N/% ^2^	63/78.5
Proteinuria (g/24 h) median (IQR)	2.6(4)
Proteinuria >3 g/d N/%	48/48
Proteinuria 1–3 g/d N/%	31/31
Proteinuria <1 g/d N/%	21/21
Active urine sediment N/%	92/92
Hypertension N/%	28/28
Serum albumin (g/dL) mean ± SD	3.1 ± 0.8
Serum Cr (mg/dL) median(IQR)	0.8(0.5)
eGFR (mL/min/1.73 m^2^) median(IQR)	94.5(50)
eGFR >60 N/%	75/75
eGFR 30–60 N/%	14/14
eGFR <30 N/%	11/11
LN class	
III N/%	28/28
IV N/%	47/47
III + V N/%	9/9
IV + V N/%	16/16
Number of crescents median(IQR)	2(4)
Activity Index median(IQR)	10(6)
Chronicity Index median(IQR)	2(2)
Interstitial fibrosis/tubular atrophy *^,3^	
<25% N/%	85/87
>25% N/%	13/13
Induction Treatment	
mycophenolic acid N/%	27/27
cyclophosphamide N/%	69/69
other	3/3
none N/%	1/1
Maintenance Treatment	
mycophenolic acid N/%	77/77
azathioprine N/%	8/8
cyclophosphamide N/%	5/5
none N/%	10/10
Duration of total treatment (months)median(IQR)	39(38)

eGFR: estimated glomerular filtration rate using the CKD-EPI formula, SLEDAI: systemic lupus erythematosus disease activity index, anti-ds DNA: antibodies against double-stranded DNA. * refers to the percentage of the renal cortex involved by interstitial fibrosis and tubular atrophy. ^1^ data available for 84/100 patients, ^2^ data available for 80/100 patients, ^3^ data available for 98/100 patients.

**Table 2 jcm-11-05017-t002:** Predictors of time to complete response.

Variables	Univariate Models		Multivariate Model	
	HR	95% Cis (*p*-Value)	HR	95% Cis (*p*-Value)
Age (years)	1.01	0.98, 1.02 (0.53)		
Sex				
Male	Reference Group			
Female	1.29	0.73, 2.29 (0.36)		
Time from SLE diagnosis to LN (years)	0.98	0.94, 1.03 (0.6)		
Hypertension				
No	Reference Group			
Yes	0.7	0.43, 1.16 (0.18)		
Low C3				
Yes	Reference Group			
No	0.73	0.40, 1.33 (0.31)		
Low C4				
Yes	Reference Group			
No	0.68	0.4, 1.15 (0.15)		
Anti-dsDNA antibodies				
Negative	Reference Group			
Positive	1.26	0.7, 2.32 (0.44)		
eGFR at diagnosis(mL/min/1.73 m^2^)				
>60	Reference Group			
<60	0.66	0.39, 1.13 (0.13)		
Proteinuria at diagnosis (g/day)				
>1.5	Reference Group			
<1.5	1.77	1.10, 2.84 (0.01)		
LN class				
III	Reference Group			
IV	0.87	0.5, 1.47 (0.62)		
III/IV + V	0.69	0.37, 1.31 (0.26)		
Number of Crescents	0.97	0.93, 1.01 (0.24)		
Activity Index	1	0.96, 1.05 (0.75)		
Chronicity Index	0.92	0.82, 1.04 (0.19)		
Interstitial fibrosis/Tubular atrophy				
<25% *	Reference Group			
>25% *	0.82	0.39, 1.72 (0.61)		
Induction treatment **				
CYC	Reference Group			
MPA	0.92	0.55, 1.56 (0.78)		

eGFR: estimated glomerular filtration rate using the CKD-EPI formula, SLEDAI: systemic lupus erythematosus disease activity index, anti-ds DNA: antibodies against double-stranded DNA, CYC: cyclophosphamide, MPA: mycophenolic acid. * refers to the percentage of the renal cortex involved by interstitial fibrosis and tubular atrophy, ** the two treatment groups were compared with and without the concomitant use of rituximab.

**Table 3 jcm-11-05017-t003:** Predictors of flares.

Variables	Univariate Models		Multivariate Model	
	OR	95% CIs (*p*-Value)	OR	95% CIs (*p*-Value)
Age (years)	0.95	0.92, 0.98 (0.02)	0.98	0.94, 1.03 (0.58)
Sex				
Male	Reference Group			
Female	0.5	0.19, 1.42 (0.2)		
Time from SLE diagnosis to LN (years)	0.93	0.84, 1.04 (0.23)		
Hypertension				
No	Reference Group			
Yes	0.96	0.37, 2.47 (0.94)		
Low C3				
Yes	Reference Group			
No	0.9	0.29, 2.96 (0.9)		
Low C4				
Yes	Reference Group			
No	0.3	0.09, 0.99 (0.05)		
Anti-dsDNA antibodies				
Negative	Reference Group			
Positive	0.96	0.29, 3.1 (0.94)		
eGFR at diagnosis (mL/min/1.73 m^2^)				
>60	Reference Group			
<60	0.7	0.27, 1.98 (0.5)		
Proteinuria at diagnosis (g/day)				
<2	Reference Group			
>2	3	1.14, 7.89 (0.02)		
LN class				
III	Reference Group			
IV	0.59	0.22, 1.55 (0.28)	0.38	0.09, 1.6 (0.19)
III/IV + V	0.21	0.05, 0.8 (0.02)	0.13	0.01, 0.8 (0.02)
Number of Crescents	0.96	0.88, 1.05 (0.44)		
Activity Index	1.01	0.91, 1.11 (0.8)		
Chronicity Index	0.77	0.59, 1.01 (0.06)		
Interstitial fibrosis/Tubular atrophy				
<25% *	Reference Group			
>25% *	0.6	0.15, 2.4 (0.48)		
Induction treatment **				
CYC	Reference Group			
MPA	0.25	0.08, 0.8 (0.02)	0.14	0.03, 0.7 (0.01)
Time to PR (months)	1.14	1.02, 1.26 (0.01)		
Time to CR (months)	1.05	1.01, 1.1 (0.01)		
Time to CR/PR (months)	1.11	1.02, 1.21 (0.01)		
Proteinuria at 12 m (g/day)				
<0.8	Reference Group			
>0.8	3.38	1.14, 10 (0.02)	4.12	1.15, 14 (0.02)

eGFR: estimated glomerular filtration rate using the CKD-EPI formula, SLEDAI: systemic lupus erythematosus disease activity index, anti-ds DNA: antibodies against double-stranded DNA, CYC: cyclophosphamide, MPA: mycophenolic acid, CR: complete response, PR: partial response. * refers to the percentage of the renal cortex involved by interstitial fibrosis and tubular atrophy, ** the two treatment groups were compared with and without the concomitant use of rituximab.

**Table 4 jcm-11-05017-t004:** Predictors of long-term adverse renal outcome (stage 3–4 CKD, ESRD).

Variables	Univariate Models		Multivariate Model	
	OR	95% CIs (*p*-Value)	OR	95% CIs (*p*-Value)
Age (years)	1	0.96, 1.03 (0.9)		
Sex				
Male	Reference Group			
Female	0.8	0.2, 2.5 (0.7)		
Time from SLE diagnosis to LN (years)	1	0.9, 1.1 (0.9)		
Hypertension				
No	Reference Group			
Yes	2.28	0.83, 6.2 (0.1)		
Low C3				
Yes	Reference Group			
No	0.75	0.2, 3 (0.68)		
Low C4				
Yes	Reference Group			
No	1.6	0.53, 4.9 (0.4)		
Anti-dsDNA antibodies				
Negative	Reference Group			
Positive	0.45	0.1, 1.5 (0.2)		
eGFR at diagnosis (mL/min/1.73 m^2^)				
>60	Reference Group			
<60	6	2, 16 (0.001)		
Proteinuria at diagnosis (g/day)				
<1.5	Reference Group			
>1.5	1.7	0.56, 5.12 (0.34)		
LN class				
III	Reference Group			
IV	0.59	0.19, 1.77 (0.34)		
III/IV + V	0.62	0.17, 2.24 (0.47)		
Number of Crescents	1	0.93, 1.07 (0.91)		
Activity Index	1.03	0.92, 1.16 (0.5)		
Chronicity Index	1.13	0.87, 1.47 (0.34)		
Interstitial fibrosis/Tubular atrophy				
<25% *	Reference Group			
>25% *	5.44	1.59, 18 (0.007)	7.7	1.48, 40 (0.01)
Induction treatment **				
CYC	Reference Group			
MPA	1.02	0.35, 3 (0.95)		
Time to PR (months)	1.14	1.04, 1.26 (0.006)		
Time to CR (months)	1.04	0.99, 1.1 (0.1)		
Time to CR/PR (months)	1.1	1.02, 1.2 (0.01)		
Proteinuria at 12 m (g/day)				
<0.8	Reference Group			
>0.8	9.5	1.84, 20 (0.003)	10.8	2.7, 42 (0.001)

eGFR: estimated glomerular filtration rate using the CKD-EPI formula, SLEDAI: systemic lupus erythematosus disease activity index, anti-ds DNA: antibodies against double-stranded DNA, CYC: cyclophosphamide, MPA: mycophenolic acid, CR: complete response, PR: partial response. * refers to the percentage of the renal cortex involved by interstitial fibrosis and tubular atrophy, ** the two treatment groups were compared with and without the concomitant use of rituximab.

## Data Availability

The data supporting this study’s findings are available from the corresponding author upon reasonable request.

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
