# Peer review of "Predictors of Early Response, Flares, and Long-Term Adverse Renal Outcomes in Proliferative Lupus Nephritis: A 100-Month Median Follow-Up of an Inception Cohort"

_jcm, 2022, doi:10.3390/jcm11175017_

Round 1

Reviewer 1 Report

This is a meticulously elaborated study from an statistical point of view that, although it probably does not add new information to the literature already published. However, I think it is interesting to publish this series of patients to provide more evidence in this regard, especially since it is a cohort very homogeneous in terms of maintenance treatment (the patients were not treated –or at least it is not mentioned– with calcineurin inhibitors or with belimumab, the latter even though the indication had been extrarenal). I would like to make some comments and suggestions:

Table 1:

-        -Since there are 100 patients, why are not equal the absolute number and percentage of C3, C4 and positive anti-ds DNA antibodies? Same with IFTA.

-        -Mixed class included all together III/IV +/- V. It is interesting to precise how many III+V and IV+V are in total specifically. This can be difficult to analyze, but two bias may exist: first, by including mixed classes in a proliferative LN cohort, we have to assume the risk that proteinuria-based analyzes might be heavily influenced by this membranous component. It would be a purest analysis to include only pure lesions (at the cost of sacrificing “n”, which is probably not recommended). Second, it is important to know how many patients were in each group, because looks like patients with mixed lesions were treated de facto as if they were class III (with MPA, etc).

Induction treatment (point 3.2):

-        - I would recommend to reflect the regimen (oral vs iv? ELNT vs NIH?)

-        -Why was used for induction rituximab in 12 pacients treated with CYC and 4 patients treated with MPA? This might have an effect in the response to the treatment, probably I would analyze if this have an impact in the subsequent analysis (determinants of response, flares etc). Was the analysis stratified?

Repeat biopsies (line 208): I recommend a brief explanation (or a table) with the results. All cases confirmed activity? Was observed other lesions that could explain the “flare”? vascular lesions, etc.

Renal flares and determinants of flares (point 3.4). Regarding treatment, it is mentioned that an induction treatment with MPA was associated with lower risk of flares. Since the median time to flare was 38 months, it would be also interesting to analyze the effect of the maintenance treatment, but probably this won’t be possible since almost all patients were on MPA and only 10% aprox with AZA.

It would be interesting to include in the text to comments:

-        -Regarding the burden of immunosupression of patients who died and their characteristics (young people with causes probably related with IS: cancer (bladder, CYC?) and sepsis.

-        -What happened, regarding renal function and survival, with those patients who not responded to treatment?

Minor recomendations:

Line 64: calcification system – classification system

Line 67-69, reedit as “time of histological diagnosis of LN; 3, 6, 9, 12, 18, 24, 36, 72 months after the diagnosis; time of renal flare (with or without a repeat biopsy), and at last follow-up visit”.

Line 158: didn’t. 225 y 227: wasn’t; 345 didn’t. Better did not and was not.

Tables are difficult to read it since they use the same format for mean +/- SD, median/IQR and N/%. I would recommend, for example: mean +/- SD, median [IQR] and N (%) or N/%

Table 1: proteinuria, I would recommend to express in median [IQR].

Supplementary table S1: I guess it is regarding induction treatment?

Reviewer 2 Report

Dear authors.

It is my pleasure to review your manuscript and collaborate with you for publishing process. You have analyzed in an inception cohort of SLE patients with renal compromise, the predictors in different scenarios of rena  outcomes. The manuscript is well written, the statistical analysis is well managed and the discussion is accordingly to the results you found. Sometimes I found that the results part have redundant information which can be seen in the suplementtary files. I advise you to review it in order to make the manuscript more concise and readers friendly. The lines described below could be improved according to my concept.

Abstract

Line 16 include de acronymous for proliferative lupus nephritis (PLN).

Key words: avoid to use the same words used in the title.

Introduction

Is there in the literature some data ( in percentage or numbers) that could be used as baseline comparators for your follow-up experience?

Methods

Line 63 . Is it possible to have a brief description of the different classes for proliferative LN in biopsy findings.

Discussion

Line 281 please use the first paragraph as the summary of the main findings according to the primary objective of the study.

Conclusion

Lines 353-355: could you be more concise and clearer  with this idea.

Lines 356-358: please be more concise and write just the findings without comparing with other data.
